# American public opinion on artificial intelligence in healthcare

**Jessica Rojahn**[1]*, **Andrea Palu**[1], **Steven Skiena**[2], **Jason J. Jones**[1,3]

**1** Department of Sociology, Stony Brook University, Stony Brook, New York, United States of America,
**2** Department of Computer Science, Stony Brook University, Stony Brook, New York, United States of America, **3** Institute for Advanced Computational Science, Stony Brook University, Stony Brook, New York, United States of America

* Jessica.rojahn@stonybrook.edu

**Data Availability Statement:** All data files are available from the OSF database (https://osf.io/gtz3q/).

**Funding:** This material is based upon work supported by the National Science Foundation

## Abstract

Billions of dollars are being invested into developing medical artificial intelligence (AI) systems and yet public opinion of AI in the medical field seems to be mixed. Although high expectations for the future of medical AI do exist in the American public, anxiety and uncertainty about what it can do and how it works is widespread. Continuing evaluation of public opinion on AI in healthcare is necessary to ensure alignment between patient attitudes and the technologies adopted. We conducted a representative-sample survey (total N = 203) to measure the trust of the American public towards medical AI. Primarily, we contrasted preferences for AI and human professionals to be medical decision-makers. Additionally, we measured expectations for the impact and use of medical AI in the future. We present four noteworthy results: (1) The general public strongly prefers human medical professionals make medical decisions, while at the same time believing they are more likely to make culturally biased decisions than AI. (2) The general public is more comfortable with a human reading their medical records than an AI, both now and "100 years from now." (3) The general public is nearly evenly split between those who would trust their own doctor to use AI and those who would not. (4) Respondents expect AI will improve medical treatment but more so in the distant future than immediately.

## Introduction

Healthcare is always evolving. Large investments in new medical technologies result in a quickly moving stream of proposed advancements in healthcare. One of those advancements is the use of artificial intelligence (AI) when making medical decisions, performing data analysis, and assisting directly with patient care [1]. Artificial intelligence can be defined as a computerized system that can perform human-like thinking and tasks such as perception, categorization, recognition, and decision-making [2]. Generally, the term AI is often used to refer to algorithms and statistical models that can communicate and reason independently in a variety of scenarios in a way that is similar or even superior to a human. AI algorithms often learn from prepared data and can perform tasks on future similar data, such as recognition, categorization, pattern inference, and threshold decision-making [3]. Medical practitioners

under grant IIS-1927227. The funders had no role in study design, data collection and analysis, decision to publish, or preparation of the manuscript. JJJ, SS IIS-1927227 National Science Foundation https://www.nsf.gov/.

seek to develop novel methods, such as using AI, to understand and solve increasingly large and complicated medical challenges in a wide variety of medical specialties, such as reducing the number of dermatology patients by helping patients identify any skin conditions that are urgent or not urgent [3–5]. The primary advantage of AI in this endeavor is the ability to learn statistical relationships within large amounts of data quickly and, as a result, make more accurate and unbiased decisions compared to humans [6].

One of the most common ways AI is already involved in healthcare is through the analysis of vast amounts of electronic medical record data [7]. Electronic medical records aim to compound patient information across medical encounters with multiple providers and update the records with each additional encounter. AI algorithms can be used to organize and check this ever-increasing amount of medical data in order to improve patient safety. For example, AI can be used in identifying and preventing potential medical errors such as prescription allergies or harmful drug interactions between new and existing prescriptions [8]. This can reduce time burdens for medical providers so that they can spend more time interacting with patients instead of cross-checking extensive paperwork. By reducing risk for patients and increasing efficiency, medical AI has been estimated to lower total healthcare spending in the US by $300 to $450 billion [1].

The benefits of medical AI can also be seen in specialty areas including wearable medical devices, mental and behavioral health, reproductive health, and surgery. For example, wearable medical electronic devices, such as smartwatches, hearing aids, and wristbands, directly collect health data over time and can enable physicians and patients to monitor long term health trends [9]. It is hoped that medical AI can efficiently monitor this constant stream of data to help medical providers make better and faster diagnosis and treatment decisions. Advanced diabetes management is one of the fastest growing examples of this with AI being used to continuously monitor a patient's glucose with minimally invasive and automatic devices [10]. By using AI in wearable glucose monitors, dangerous health events can be detected more quickly and accurately, and possible future events could even be predicted.

Other AI tools, such as online chat bots, are also increasingly being used in the treatment of mental and behavioral health [11, 12]. AI chat bots, for example, can enable patients to receive faster and direct feedback on their own, which can help in a crisis or prepare them for future medical consultations and treatments. AI algorithms can also be used to detect possible concerns in written text by analyzing the language in extremely large amounts of text and detecting possible mental health concerns for a patient or even large groups [12]. For example, by using text-based emotion AI, Deshpande and Rao [13] were able to detect possible signs of depression in users by analyzing thousands of tweets. Luxton [14] reviewed the many ways AI could be used in mental and behavioral healthcare including using virtual avatars to talk with patients and gather information on symptomology, developing augmented and virtual reality tools, and creating therapeutic computer games.

AI is also having a direct impact on how medical providers are able to perform complicated medical procedures such as artificial reproduction and surgical procedures. In the quickly developing field of artificial reproductive technologies (ART), AI systems can identify and predict which cells will result in a higher likelihood of a successful embryo [15]. AI can also assist with gamete and embryo preservation, the fertilization process itself, and other procedures such as genetic testing. It is hoped these improvements can lower the extreme cost of ART procedures and reduce the emotional and physical strain of using ART by increasing the likelihood of a successful pregnancy and birth.

In the operating room, medical AI can be used to enable surgeries to be less invasive, resulting in less risk for the patient and easier recovery [4]. With AI support, surgeons can focus on the most useful data collected before and during surgery so they can make more accurate and

efficient decisions regarding pre-op, post-op care, and the surgery itself [16]. Current surgical techniques are constantly being improved and new techniques are being developed through use of surgical robotics and AI [12].

Development of medical AI even goes beyond the doctor's office into active military use. The US military is looking to improve centuries-old triage procedures by investing in the development of AI algorithms [17]. The Defense Advanced Research Projects Agency (DARPA) is developing AI systems to replace human judgements to make triage decisions in quickly changing and stressful emergencies. The hope is to remove human bias and make more effective decisions in order to save lives. However, there is concern these algorithms will be created with biased input data or be unable to react acceptably in novel, unpredictable situations for which previous data do not exist. Additionally, some have raised the issue that AI decision-makers have ambiguous moral culpability when the decisions they make result in human death. In such situations, many people want to know that the person responsible for their loved one's death feels regret or did their best, but if an AI was responsible it can be seen as cold, unfeeling calculus. In addition to creating a system to efficiently make decisions, developers of medical triage AI for military deployment will need to consider how the systems will be accepted and interpreted by the subjects of the decisions made by the AI.

It is believed that as the development of AI systems improve through the collection of better and larger amounts of data as well as engineering improvements, AI will be able to make healthcare more convenient, accurate, efficient, and personalized in the future [18]. With many medical providers hoping for such improvements, it is important to understand how *patients* actually perceive and trust AI being involved with their medical care. If patients are not willing to entrust their healthcare decisions and medical records to AI systems, the implementation of medical AI will grind to a halt. However, much of the past research investigating attitudes towards medical AI has focused on the users of the systems instead of the recipients of care.

As AI is increasingly being implemented in their career field, medical professionals have the unique opportunity to see the potential benefits and drawbacks of using such technology. Among medical professionals there is a general belief that there are at least some benefits to using medical AI [19]. One recognized benefit is that AI could help reduce errors such as with drug interactions and data tracking. Those working in medical lab environments believe there can also be benefits unique for their work such as increasing test performance and helping to ensure the objectivity of their procedures and interpretations. Patient care, data analysis, scientific research, clerical work, and test results analyses are also believed to potentially be improved through the speed, accuracy, and efficiency of algorithms [20].

On the other hand, medical professionals still express some doubt and concern about the encroachment of AI into their work [21]. There is concern AI will not be able to complete tasks that require typically human skills such as judgement, understanding, and decision-making [19]. AI, as a programmed algorithm, is seen as less capable of providing humanistic patient care due to not actually being a human with emotions and the ability to understand the human experience. The objectivity of AI, sometimes seen often as a strength, is also seen as a flaw that prevents it from being able to provide quality empathic care, gather all possible and accurate information from patients who may be scared or reluctant, or treat each patient in a unique manner specific to their individual needs [22]. As such, there is strong belief that human physician mediated care will always be preferred by the majority of patients and would lead to the best possible health outcomes with AI as a useful support tool [22]. In general, medical professionals tend to believe that AI could reduce errors and increase efficiency, but it will never be able to take over human medical jobs completely nor perform at a level equal or superior to those of human [20, 23].

As demonstrated, most of the past research about attitudes towards medical AI has been done with professionals and students who are working in the medical field, however, this is only half of the equation. The current study redirects the focus to the attitudes of the general public in the United States towards AI in medicine. In other words, how do the patients who would be on the receiving end of AI-directed medical care feel about its involvement?

Although there is great variability in the characteristics and medical experiences of the general public, there are some common themes in their opinion towards medical AI and most of them can be described as distrusting or anxious. In one study, after using an AI device to determine their medical diagnosis, patients reported concern about communication barriers with the AI as well as feelings of unease and a lack of trust regarding its performance, mechanisms, and unregulated standards [24]. The mistrust of medical AI systems comes from feelings about the system itself as well as the technology companies developing the programs. Respondents voice concern about data privacy, technical issues with gathering high quality and accurate medical data, and technology companies prioritizing profitability of their business over human lives [25, 26].

AI is often thought of as a "black-box" with no possible way for laymen to understand how the system came up with its output [27]. It is this lack of understanding that could be a major cause of decreased trust in AI as those without background knowledge of AI cannot make educated assumptions about the system they need to trust. Trust has been proposed to be a multipart concept including reliability, competence, and intentions, and without the ability to know how AI systems are created and function, perceptions of all three dimensions are unlikely to be very high [28]. Although the human mind can also be considered a sort of "black-box," most people express less trouble trusting their human doctors.

To compare, Juravle et al. [29] investigated the nuances of the public's trust of medical AI with a multi-stage experiment using hypothetical scenarios that alternated between a human or AI doctor providing a first and second diagnosis. They found significantly more trust for human doctors over AI. When the first diagnosis was from an AI and confirmed by a human, the trust in the diagnosis was increased, but when an AI confirmed the human doctor's diagnosis trust remained relatively unaffected. Even when participants were informed that the AI outperformed the human doctor, their trust in the AI was relatively unchanged. The trust that patients have in their medical caregiver is extremely important to their health outcomes as higher trust has repeatedly been found to correlate positively with following medical advice, complying with medication and treatment plans, and comfort with sharing potentially relevant personal medical information [30]. Juravle et al.'s [29] study regarding trust in medical AI further supports this claim as participants reported a higher probability that they would follow the suggested medical treatment if it came from a human doctor versus an AI.

Some research about public attitudes also supports the main concern about medical AI that doctors have—specifically that due to AI's 'lack of enthusiasm' it will never be able to provide humanistic care on its own as compared with human physicians [25]. This likely contributes to the strong preference for a human doctor being responsible for medical care over an AI alone [26].

However, there is potential for the use of virtual medical providers for more informational tasks, such as explaining treatment plans and updated medical information to patients [31]. For example, hospital discharge could particularly benefit from the efficiency of AI as various medical errors that result in patients being re-hospitalized later with complications can occur if the information is not adequately explained to patients. Medical AI could reduce the time patients need to wait for their information and enable patients to repeatedly ask questions they may be uncomfortable or unwilling to ask a busy human professional [31]. In an effort to improve this experience, virtual programs are being developed with a focus on interactivity

and relational behavior with the specific goal of reducing the burdens on medical staff and to better educate patients about their healthcare.

Bickmore, Pfeifer, and Jack [31] have conducted studies to evaluate virtual nurse programs as potential nurse substitutes or tools to explain the discharge steps and healthcare plans to patients. With a virtual nurse program, patients are able to go over their information at a comfortable pace which enables them to ask more questions and get clarifications. Many patients describe feeling ignored, talked down to, or dismissed by overworked and busy medical professionals, especially in a hospital setting; virtual programs might allay patients' negative feelings by being constantly available and emotionally neutral. When asked to compare a virtual nurse to human providers for discharge, patients who used the program reported feeling they were getting the one-on-one attention, time, and concern they needed but did not experience with human providers. In their study, over one-third of patients preferred the virtual nurse for discharge with only 26% of patients indicating they would prefer a human [31].

Bickmore, Pfeifer, and Jack's [31] virtual nurse program is also an excellent example of how AI can be used in tandem with human staff to reap the benefits of both. In this case, the system is set up for patients to interact with the virtual nurse first, but a human nurse can still follow-up with the patient before final discharge to confirm they understand their medical information accurately. Using a virtual program can also help patients become empowered and prepared for speaking with the human medical staff. For example, after consulting with the virtual nurse who cannot feel tired, annoyed, or overworked, patients reported either having their questions already answered or feeling more prepared to ask their questions to human providers with a greater chance of understanding and success [31].

Although the public tends to be wary of AI being involved in their medical care, they are still hopeful that with more development and improvements AI will become more trustworthy [24, 32]. This largely comes from the belief that AI can be unbiased and honest about its analyses and decisions regardless of the patient's social class or characteristics that have a history of discrimination in healthcare. The public also tends to perceive AI to be more objective, convenient, efficient, and cheaper [26]. Interestingly, these perceived benefits of medical AI are also seen as potential risks—especially when the AI is self-sufficient and without a human directly involved. For now, much of the public feels that AI is too new to entirely trust with their healthcare and see it as a supplemental tool that can support but not replace human doctors [25–26, 29].

It is important to evaluate the public's trust in medical AI. In order to implement AI in the medical field with the highest chances of success, positive health outcomes, and increased satisfaction, practitioners should understand patients' attitudes and beliefs. Toward that end, the current studied used an online survey platform with a quickly accessible nationally representative U.S. sample to assess the attitudes of the general public towards medical AI.

The hypotheses for this project were that (1) the American public would be more trusting of human medical professionals over AI in regard to healthcare decision-making and privacy but that (2) the American public would still express hope that AI will improve healthcare in the future.

## Method

### Data collection and survey design

A survey was created and distributed to respondents through the online survey platform Google Surveys in March 2021 [33]. Google Surveys was designed to enable survey creators to measure attitudes and opinions of a target audience in a fast and cost-effective way [34]. Google Surveys provides access to nationally representative samples—in this case of the general American public–and includes gender, age, and region demographics. Post-stratification weights for

computing population estimates (based on comparison of the sample and American Community Survey data) are also included.

## Ethics statement

Respondents consented by accepting and completing the online survey. They could opt out at any time without penalty. The identity of the respondents cannot readily be ascertained, directly or through identifiers linked to the subjects. The survey data collection qualifies as Exempt research under the criteria specified in Department of Health and Human Services code § 46.104(d)(2).

A 9-item survey designed to measure attitudes and opinions about Artificial Intelligence in the medical field was sent to Google Play app users. Users earned Google Play credit for participating. Half the respondents received Survey Form A and the other half received Form B. All questions on Survey Form A and Form B are identical except for Question 8. Question 8 took the form of a survey experiment in which half the respondents were asked "In your opinion, how much will artificial intelligence improve treatment of medical problems over the current status quo in the next 10 years?" The other half of respondents were prompted with 50 years instead of 10. Response options for this question included three categories: "Not at all", "Somewhat", and "A great deal."

Two items assessed preferences for who should make medical decisions in different scenarios. One item asked about triage decisions in an emergency room with triage defined as "which patients should be treated first." The other item asked about discharge decisions with discharge defined as "should this patient leave the hospital now." There were two response options for each item: "a human medical professional (e.g., a doctor)" or "a computer algorithm (e.g., an AI system)."

One item asked respondents to indicate whether a human medical professional or a computer algorithm was more likely to make culturally biased decisions. To counter any affect from question order, this item preceded the triage and discharge items on Form B, but followed the triage and discharge items on Form A.

Two items investigated how comfortable respondents were with an artificial intelligence (AI) computer system reading their medical records now and 100 years from now. This was measured with a 7-point Likert scale with responses ranging from "not comfortable at all" (1) to "very comfortable" (7). For comparison, two additional items asked respondents to report how comfortable they were with human doctors other than their own reading their medical records now and 100 years from now. In sum, these four items had the effect of providing a 2x2 within-subject survey experiment. One independent variable was Reader, with levels "AI computer system" and "human doctors other than your own." The other independent variable was Time, with levels "now" and "100 years from now."

One item was included to assess respondents' attitudes toward human doctors and AI working together. Respondents indicated either "Yes" or "No" to the question "Would you trust your own doctor to use an artificial intelligence system to diagnose a condition for you?"

See Table 1 for the full, exact item text, response options, and item order on each form.

## Respondent sample

Demographic data included gender, age, and region in the United States. In our sample, male respondents (55% of the sample) were over-represented. Adults 55 and older were under-represented (12% of the sample). The sample was well-distributed geographically with each of the regions (Northeast, Midwest, West, South) accounting for 23% to 28% of the sample each. See Table 2 for complete sample demographic information. All respondents were over the age of

**Table 1. Survey questions and response options with survey form question order.**

| Form A (Form B) | Question Text | Responses |
|---|---|---|
| 1 (2) | When an emergency room must make a triage decision—which patients should be treated first—which system do you think should be used? | A computer algorithm (e.g an AI system)<br>A human medical professional (e.g. a doctor) |
| 2 (3) | When it comes to an individual patient's discharge decision—should this patient leave the hospital now—which system do you think should be used? | A computer algorithm (e.g an AI system)<br>A human medical professional (e.g. a doctor) |
| 3 (1) | In your opinion, which system is more likely to make culturally biased decisions? | A computer algorithm (e.g an AI system)<br>A human medical professional (e.g. a doctor) |
| 4 (7) | For the purpose of greater scientific understanding, how comfortable are you with *human doctors* (other than your own) reading your medical records *now*? | 1 (Not At All Comfortable)<br>7 (Very Comfortable) |
| 5 (6) | For the purpose of greater scientific understanding, how comfortable are you with an *artificial intelligence (AI) computer system* reading your medical records *now*? | 1 (Not At All Comfortable)<br>7 (Very Comfortable) |
| 6 (5) | For the purpose of greater scientific understanding, how comfortable are you with *human doctors* (other than your own) reading your medical records *100 years from now*? | 1 (Not At All Comfortable)<br>7 (Very Comfortable) |
| 7 (4) | For the purpose of greater scientific understanding, how comfortable are you with an *AI computer system* reading your medical records *100 years from now*? | 1 (Not At All Comfortable)<br>7 (Very Comfortable) |
| 8 (8) | In your opinion, how much will artificial intelligence improve treatment of medical problems over the current status quo in the next 10 (50) years? [a] | Not At All<br>Somewhat<br>A Great Deal |
| 9 (9) | Would you trust your own doctor to use an artificial intelligence system to diagnose a condition for you? | Yes<br>No |

[a] Question 8 on Survey Form B asked about 50 years.

**Table 2. Sample demographics (N = 203).**

| | Demographics | n (%) |
|---|---|---|
| Age | | |
| | 18–24 | 39 (19%) |
| | 25–34 | 58 (29%) |
| | 35–44 | 51 (25%) |
| | 45–54 | 30 (15%) |
| | 55–64 | 11 (5%) |
| | 65 + | 14 (7%) |
| Gender | | |
| | Male | 112 (55%) |
| | Female | 91 (45%) |
| Geographic Region | | |
| | US—Midwest | 49 (24%) |
| | US—Northeast | 47 (23%) |
| | US—South | 58 (28%) |
| | US—West | 49 (24%) |

18. Two hundred and six total responses were collected. Only participants who completely answered all demographic and survey items were included in analysis, resulting in an N of 203: 99 respondents with Survey Form A and 104 with Survey Form B.

## Results

The Google Surveys platform provided post-stratification weights such that the sample could be used to compute population estimates for census-matched United States age, gender, and region/state distributions. All statistical analyses were conducted using the R Survey package [35]. Data and analysis code are publicly available at https://osf.io/gtz3q/.

### Medical decisions and bias

When asked whether a human medical professional or artificial intelligence system should make triage and discharge decisions, the American public has a clear preference, as seen in Figs 1 and 2, $X^2_{triage}$ (1, N = 203) = 147.45, p < 0.001 and $X^2_{discharge}$ (1, N = 203) = 116.5, p < 0.001. 92.6% of the American public prefer a human medical professional over an AI to make triage decisions and 87.9% prefer similarly for discharge decisions. However, as shown in Fig 3, the American public also believes that the human providers they prefer are also more likely to make culturally biased medical decisions compared to AI, $X^2_{bias}$ (1, N = 203) = 34.043, p < 0.001.

### Comfort with others reading one's medical records

Respondents were asked four questions designed to measure how comfortable they were with one of two types of readers reviewing their medical records at two different times. Each

*When an emergency room must make a triage decision - which patients should be treated first - which system do you think should be used?*

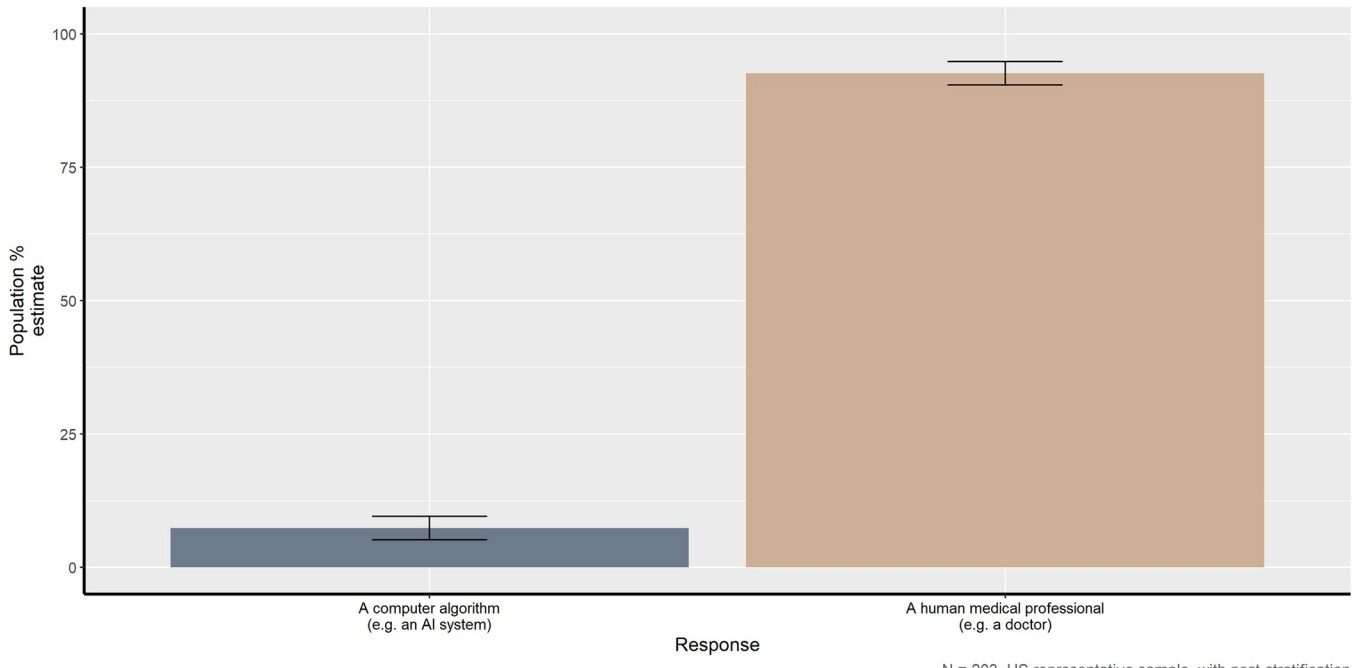

**Fig 1. When an emergency room must make a triage decision—which patients should be treated first—which system do you think should be used?**

*When it comes to an individual patient's discharge decision - should this patient leave the hospital now - which system do you think should be used?*

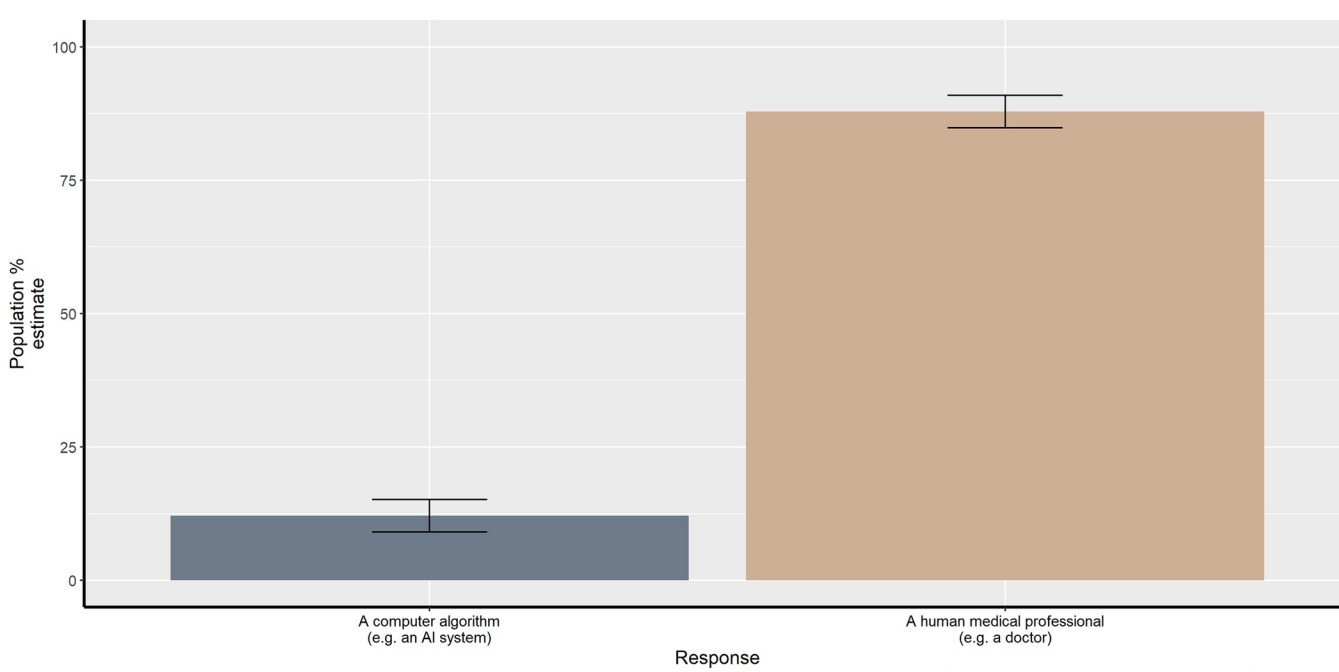

N = 203, US representative sample, with post-stratification

**Fig 2. When it comes to an individual patient's discharge decision—should this patient leave the hospital now—which system do you think should be used?**

question combined two variables: the type of reader–human doctors other than their own or an AI computer system–and the time of reading–now or 100 years from now. A repeated measures ANOVA was used to test for significant main effects and interactions of reader type and time. Fig 4 plots the results.

Both the type of reader and the time of reading had significant main effects on the comfort of the respondents. For the type of reader, respondents reported being more comfortable with human doctors reading their medical records compared to an AI system, $F(1,202) = 83.199$, $p < 0.001$, $\eta^2 = 0.099$. For the timing, respondents were more comfortable with their medical records being read 100 years from now compared to being read now, $F(1,202) = 60.579$, $p < 0.001$, $\eta^2 = 0.038$. Although the interaction of type of reader and time of reading was also found to be significant, it explained less than 1% of the variance, $F(1,202) = 9.747$, $p = 0.002$, $\eta^2 = 0.003$.

To explore the generality of these results across demographics, we conducted repeated measures ANOVAs split by gender and age. The same patterns were found. These analyses can be found in S1 Fig.

## Expectations for AI

The one item that differed between survey forms asked participants to indicate how much they believe AI will improve medical treatment in the next 10 years (Survey Form A) or the next 50 years (Survey Form B). As seen in Fig 5, the American public has significantly more hope for the impact of AI on medical treatment when thinking further out into the future compared to the more immediate future, $X^2 (1, N = 203) = 11.691$, $p < 0.01$.

*In your opinion, which system is more likely to make culturally biased decisions?*

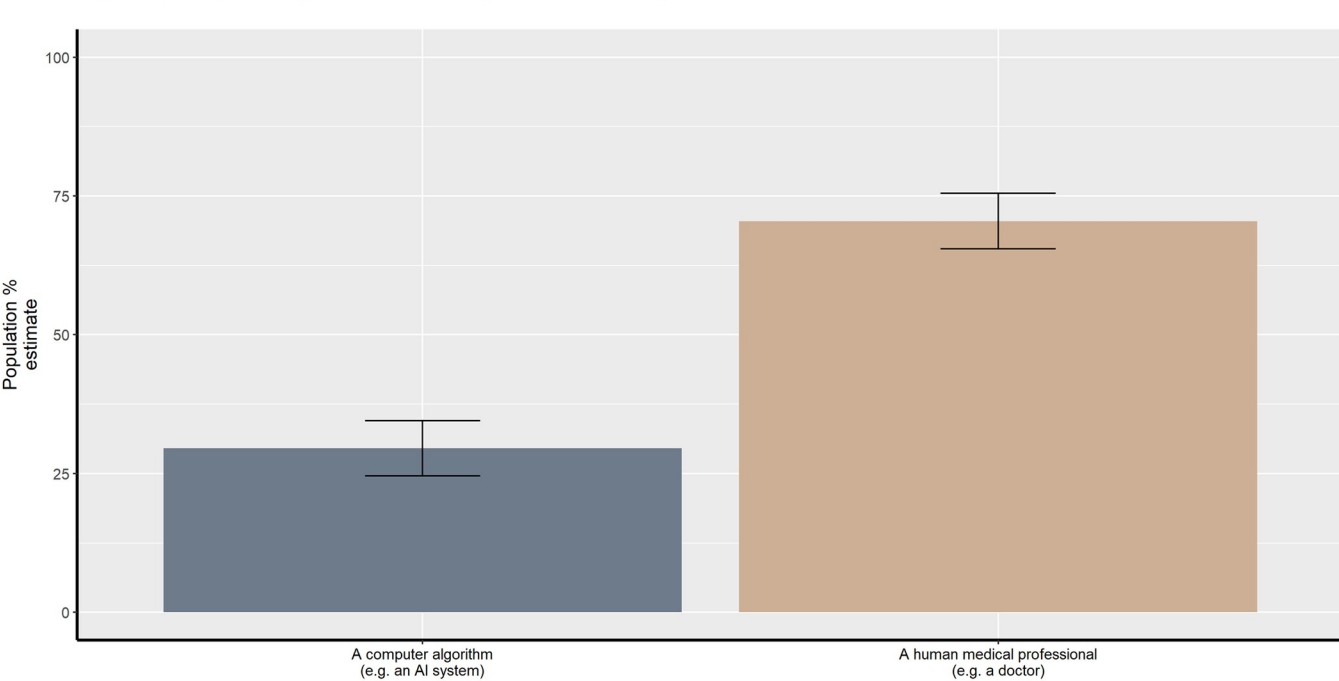

N = 203, US representative sample, with post-stratification

**Fig 3. In your opinion, which system is more likely to make culturally biased decisions?**

*For the purpose of greater scientific understanding, how comfortable are you with <Reader> reading your medical records <Time> ?*

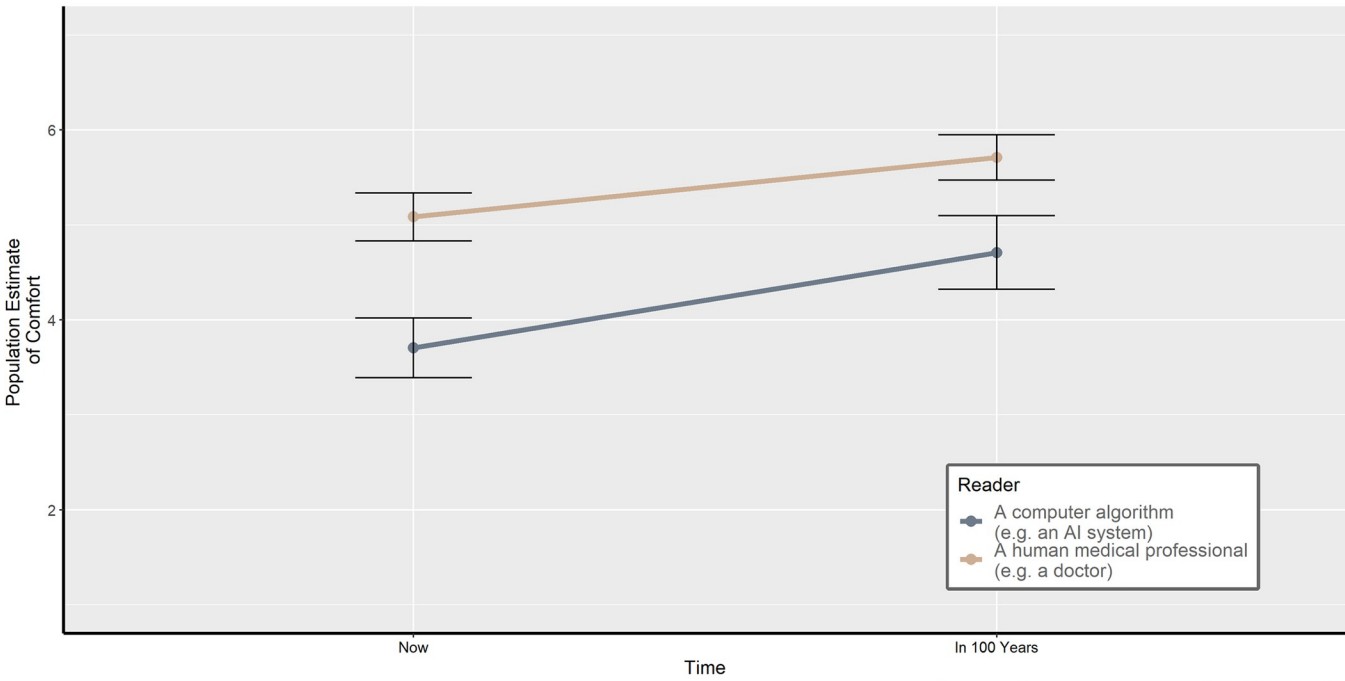

N = 203, US representative sample, with post-stratification

**Fig 4. For the purpose of greater scientific understanding, how comfortable are you with <Reader> reading your medical records <Time>?**

*In your opinion, how much will artificial intelligence improve treatment of medical problems over the current status quo in the next 10 (50) years?*

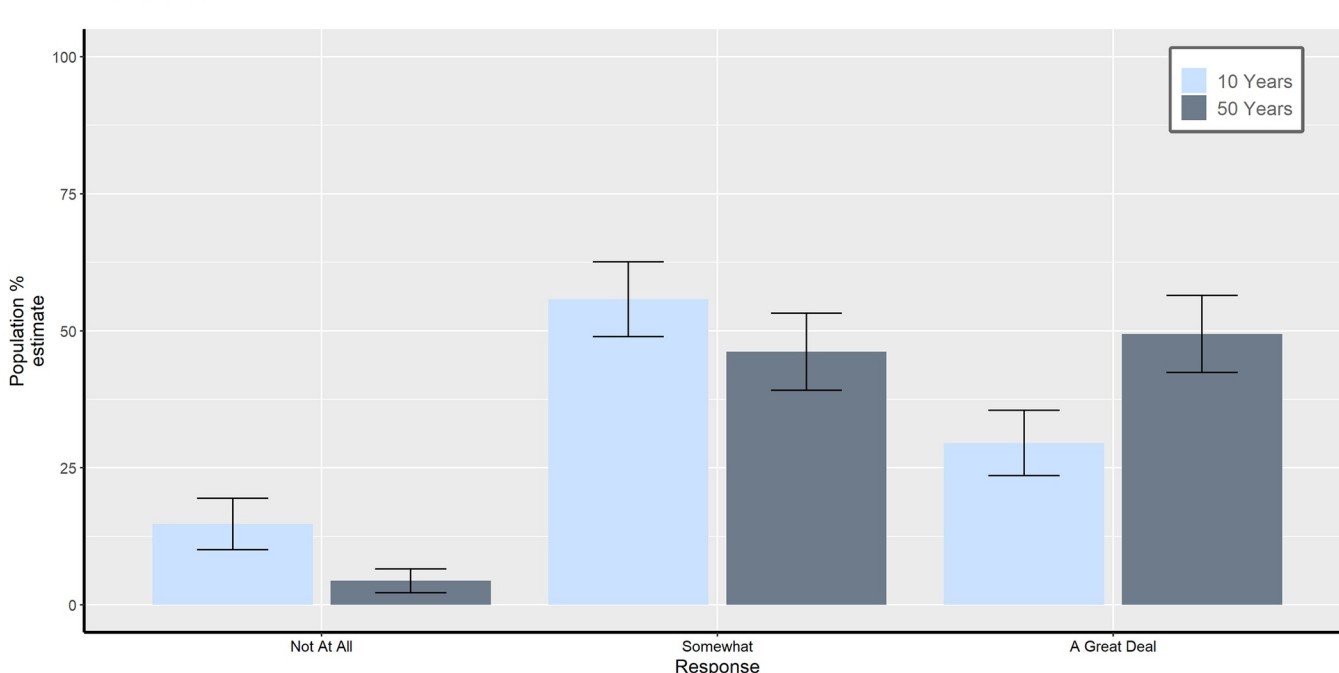

N = 203, US representative sample, with post-stratification

**Fig 5. In your opinion, how much will artificial intelligence improve treatment of medical problems over the current status quo in the next 10 (50) years?**

Analyses suggest that when thinking about how much AI may improve medical treatment in 50 years, significantly more people believe that AI will improve medicine a great deal compared to when thinking about the amount of improvement only in the next 10 years, $X^2$ (1, $N = 203$) = 7.577, $p < 0.01$. On the other side, when asked to think 10 years in the future compared to those thinking further along, significantly more respondents believe that AI will improve medical treatment "not at all", $X^2$ (1, $N = 203$) = 5.168, $p < 0.05$.

### AI as a doctor's tool

When asked if they would trust their own doctor to use an AI to diagnose a condition for them, respondents split nearly evenly between the affirmative and the negative. As seen in Fig 6, the proportion of respondents indicating they did trust their doctor did not significantly differ from those indicating that they did not trust their doctor in this case. We compared subsets of the sample and found the near-even split within men-only, women-only, and younger-only and older-only subsamples (see S2 Fig).

### Discussion

This study furthers the understanding of how patients perceive and trust AI in healthcare. The results provide four noteworthy conclusions: (1) The general public strongly prefers human medical professionals make medical decisions, while at the same time believing they are more likely to make culturally biased decisions than AI. (2) The general public is more comfortable with a human reading their medical records than an AI, both now and "100 years from now." (3) The general public is nearly evenly split between those who would trust their own doctor to

*Would you trust your own doctor to use an artificial intelligence system to diagnose a condition for you?*

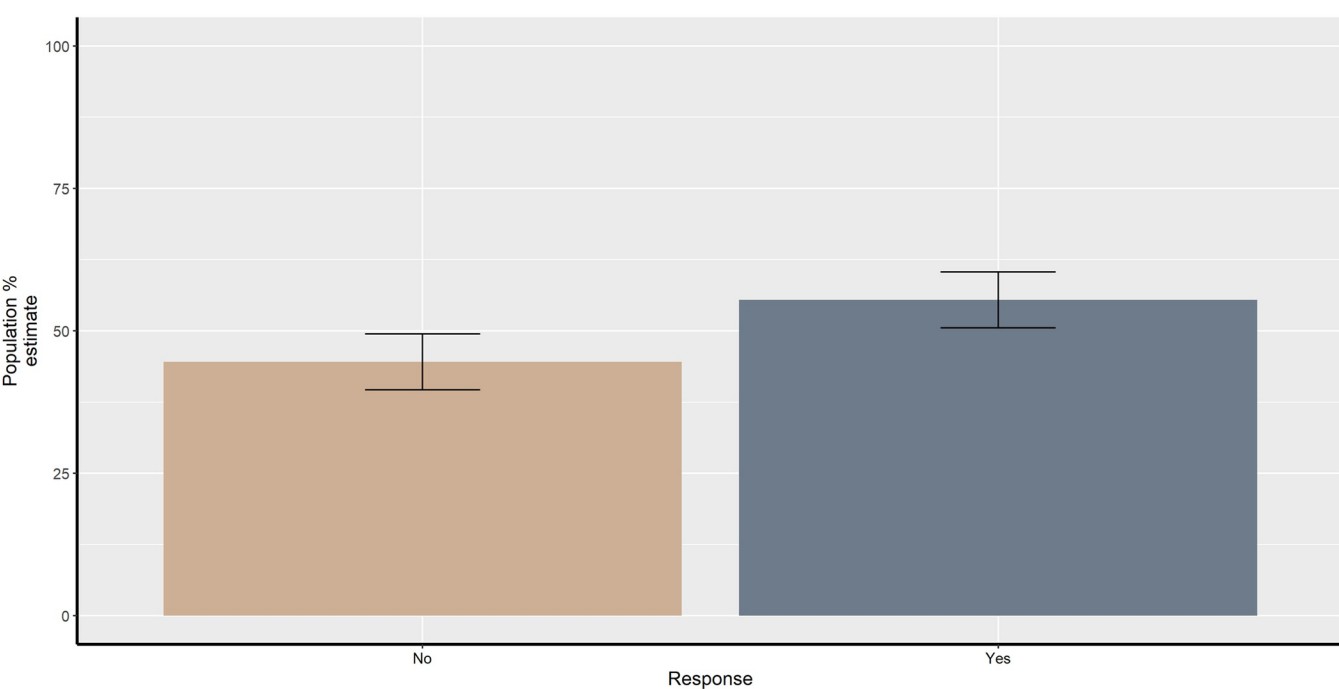

N = 203, US representative sample, with post-stratification

**Fig 6. Would you trust your own doctor to use an artificial intelligence system to diagnose a condition for you?**

use AI and those who would not. (4) The general public expects that AI will improve medical treatment—but more so in the distant future than immediately.

In Figs 1 and 2, the American public's preference for who should make medical triage and discharge decisions is quite clear and consistent. Although having an AI assist with such decisions could decrease wait times and provide patients with additional, personal attention, in both circumstances, the public significantly prefers a human physician over an AI system. This study's results are not completely surprising as they fit with a majority of past research on patient opinions; human physicians are typically trusted and followed more than an AI system or algorithm [29, 31]. The public's attitudes towards medical AI becomes more complicated, however, when the belief of which potential medical caretaker is most likely to make culturally biased decisions is taken into account.

Fig 3 clearly demonstrates that even though the general public strongly prefers the human physician making the medical decisions, they believe the human physician is more likely to make culturally biased decisions. Although it seems illogical to prefer the caretaker believed to be the most culturally biased, it may be due to greater familiarity with human physicians and the patient-doctor relationship. The patient-doctor relationship is built through dynamic interactions between both patient and doctor is often characterized by knowledge, familiarity, trust, and loyalty [36]. Patients develop familiarity with their doctor through their regular personal experiences where they feel the doctor is interested in them and cares about them. They tend to start off automatically trusting their doctor, and more positive experiences deepen that trust. Loyalty is also important as patients will prefer the same doctor and doctors will ensure the patient's wishes and needs are of highest priority. The loyalty that results from the patient-doctor relationship allows for some of the doctor's medical mistakes to be forgiven, but an AI would not have that benefit and any AI mistakes would likely be more costly [28]. Therefore,

even though patients believe human doctors to be more likely to be culturally biased, their relationship with their doctors may enable them to look past this.

In other words, the public seems to prefer the 'human element' in their medical care, the interpersonal relationship, even when the 'human element' comes with humanistic flaws as well [25, 37]. AI is still thought of as a "black-box" and is perceived as in technological infancy with no way for the patient to adequately understand how it works and if they should trust it [27]. Perhaps as patients are exposed to medical AI more and it becomes more familiar, the trust in human physicians will not be enough for them to ignore their belief that humans are more likely to be biased in their treatment [24, 25].

Another possible explanation for such a clear and persistent preference for human medical experts over AI, even when the human is perceived to be more biased, could be the phenomenon referred to as algorithmic aversion [38]. Algorithmic aversion is a subjective, systematic, and biased assessment of an algorithm that is not based on unbiased observation or actual experience. In other words, having a negative opinion about algorithms without any objective reason. It can manifest in negative and distorted perceptions of algorithms that are separate from objective reality or direct experience of the AI performing worse than human agents. Although it would be logical to predict that patients would feel algorithmic appreciation [39], or a preference for AI if it is the least biased source of medical care, when they believe the alternative to be more likely to be biased, our results do not support this. In the case of medical AI, patients still prefer a human doctor over an AI even though they believe the human doctor to make more errors due to bias. Perhaps this is because when a medical AI or human doctor makes a biased error, they perceive the error to be less severe when done by a human as they believe that human to be able to learn from their mistakes compared to a pre-programmed medical algorithm [40].

The preference for human doctors over AI also extends beyond active medical decisions to reading medical records. Results repeatedly show that it does not matter if their medical records are read now or far in the future, respondents still prefer a human doctor doing the reading compared to an AI system. It is possible that familiarity with how their doctors respect them as patients and with their motivations as human caregivers, along with their anxiety about AI, is strong enough that patients would prefer to stick with the problematic, biased system and readers they are accustomed to [36]. Although it stands to reason patients prefer their records to be read in 100 years compared to now because they likely would no longer be alive to suffer the consequences of any bias or error from the reader, their discomfort with AI still keeps the human doctor as their preference.

The evidence suggests Americans are nearly evenly split between those who would trust their own doctor to use AI and those who would not, but it is not clear why. Future research will need to tease apart the different aspects present in the question to see what really going on in the minds of the public in this scenario. Further investigation and follow-up questions may determine if the lack of significant preference here is due to the strong preference for human medical providers or the distrust of AI systems.

The results of this study are consistent with past research. Generally, the American public still prefers a human physician to be responsible for their medical care over an AI. However, the attitudes of the general public seem mixed when the survey results are taken as a whole. For instance, although the preference for humans over AI was significant, AI was not seen as hopeless, useless, or completely untrustworthy compared to human caretakers. As our results demonstrate, there is hope for the future of AI in healthcare and expectation of improvement, albeit not in the immediate future. Fig 5 reveals a distinct difference in how much improvement the general public expects in the future of medical care over the current status quo as a result of incorporating medical AI in the next few decades. A greater proportion of the public believes that AI will improve medical treatment a great deal 50 years from now compared to in

only 10 years. This result demonstrates that although the American public is hopeful that AI will improve the medical field, they do not expect this improvement any time soon. This result supports past research that found AI is still seen as too new with too many questions that still need answered and too many technological issues to work through [24, 25, 32].

## Future research

Future research can be designed to follow up on the relationships revealed in this study and improve upon some of its limitations. Although the survey data was analyzed using nationally representative sample weights, a larger sample would still improve the precision of the population estimates and improve the age distribution of the sample by including more older adults. This would be important to understanding age and generational differences in attitudes towards AI being involved in their medical treatment. For example, research shows that older generations are more skeptical of healthcare technologies, have more difficulty learning to use it resulting in a stronger preference for other forms of healthcare communication and treatment [41]. As younger generations have more experience with technology and more developed AI, attitudes towards medical AI may change over time due to generational replacement.

Future research could also investigate the nuances and detailed reasons for why the public feels as they do towards medical AI. In this study, the restrictions of the Google Survey platform did not allow for follow-up questions. Future research could address this shortcoming with a more extensive set of items. Follow-up questions could also allow for more certainty regarding how participants understood each of the survey questions. For example, participants may interpret the framing of the questions, such as "the purposes of greater scientific understanding" or "in 100 years," differently. Research shows how respondents can interpret even seemingly straightforward survey items differently so perhaps some respondents in this study believe "in 100 years" means after their death (as the authors intended) while some believe it is a hypothetical situation where they were alive 100 years from now [42].

Future research should also investigate if the public's attitudes towards medical AI would change for different types of medical tasks, specialty areas, medical urgency, and risk. The current study focused on attitudes towards medical AI in general, but research shows that who the decision maker is does not matter as much as the positive or negative nature of the outcome for resource allocation decisions [43]. Future research could also investigate if the urgency and risk of a medical crisis may also have an impact on public attitudes towards medical AI. For example, during the COVID-19 pandemic, accurate and safe diagnosis was a national-level crisis. AI algorithms are able help through contact tracing apps to automatically identify if a patient is in respiratory distress along with other COVID-19 warning signs [44]. Medical AI has been found to have high accuracy in diagnosing COVID-19 and robots with AI can help with patient assessment and treatment to reduce the spread of infection. In such global and contagious crises, public attitudes towards AI being a part of healthcare may shift as the benefits become more pronounced and lifesaving.

## Conclusion

With the field of medical artificial intelligence expanding rapidly, it is more important than ever to understand how the American public feels about AI. Although the general outlook of the public is distrusting and uncomfortable with medical AI, respondents expressed hope for the future and an expectation that medical treatment will be greatly improved over the next few decades. In preparation for such a hopeful and improved future, research must investigate and explore the reasons for why the public feels as it does so that AI can become a successful and welcomed tool in health care.

## Supporting information

**S1 Fig. For the purpose of greater scientific understanding, how comfortable are you with <Reader> reading your medical records <Time>?** A computer algorithm (e.g. an AI system). A human medical professional (e.g. a doctor). By gender.
(TIF)

**S2 Fig. For the purpose of greater scientific understanding, how comfortable are you with <Reader> reading your medical records <Time>?** A computer algorithm (e.g. an AI system). A human medical professional (e.g. a doctor). By age.
(TIF)

## Author Contributions

**Conceptualization:** Jason J. Jones.

**Data curation:** Andrea Palu.

**Formal analysis:** Jessica Rojahn.

**Funding acquisition:** Steven Skiena, Jason J. Jones.

**Methodology:** Jessica Rojahn, Andrea Palu, Jason J. Jones.

**Project administration:** Steven Skiena.

**Resources:** Steven Skiena.

**Supervision:** Jason J. Jones.

**Visualization:** Jessica Rojahn.

**Writing – original draft:** Jessica Rojahn, Andrea Palu.

**Writing – review & editing:** Jessica Rojahn, Jason J. Jones.

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
