## [Decision Letter · Decision Letter 0]

25 Jan 2023

PONE-D-23-00691American Public Opinion on Artificial Intelligence in HealthcarePLOS ONE

Dear Dr. Rojahn,

Thank you for submitting your manuscript to PLOS ONE. After careful consideration, we feel that it has merit but does not fully meet PLOS ONE’s publication criteria as it currently stands. Therefore, we invite you to submit a revised version of the manuscript that addresses the points raised during the review process.

We look forward to receiving your revised manuscript.

Kind regards,

Luigi Lavorgna

Academic Editor

PLOS ONE

Journal Requirements:

Reviewers' comments:

Reviewer's Responses to Questions

**Comments to the Author**

1. Is the manuscript technically sound, and do the data support the conclusions?

Reviewer #1: Yes

2. Has the statistical analysis been performed appropriately and rigorously? 

Reviewer #1: Yes

3. Have the authors made all data underlying the findings in their manuscript fully available?

Reviewer #1: Yes

4. Is the manuscript presented in an intelligible fashion and written in standard English?

Reviewer #1: Yes

5. Review Comments to the Author

Reviewer #1: Job well done. I just have a few suggestions to improve your already good work.

page 3 - Introduction section: please provide a more complete definition of artificial intelligence e provide an example to how it is applicable to different field of medicine ( Lavorgna L, Bonavita S. Artificial intelligence will change MS care within the next 10 years: Commentary. Mult Scler. 2022 Dec;28(14):2175-2176. doi: 10.1177/13524585221133537. PMID: 36408580.)

6. PLOS authors have the option to publish the peer review history of their article (what does this mean?). If published, this will include your full peer review and any attached files.

Reviewer #1: No

---

## [Author Response · Author response to Decision Letter 0]

31 Mar 2023

page 3 - Introduction section: please provide a more complete definition of artificial intelligence e provide an example to how it is applicable to different field of medicine ( Lavorgna L, Bonavita S. Artificial intelligence will change MS care within the next 10 years: Commentary. Mult Scler. 2022 Dec;28(14):2175-2176. doi: 10.1177/13524585221133537. PMID: 36408580.)

Please find on page 3 an expanded definition of artificial intelligence. Thank you for the suggested manuscript. Unfortunately, our library was not able to obtain it, and we did not receive it after an email and ResearchGate request to the corresponding author. We have included an example of use of AI in dermatology in the revised manuscript.

We have ensured the manuscript meets PLOS ONE's style requirements.

The study did not involve minors.

We have added the following text to the Method section: Respondents consented by accepting and completing the online survey. They could opt out at any time without penalty. The identity of the respondents cannot readily be ascertained, directly or through identifiers linked to the subjects. The survey data collection qualifies as Exempt research under the criteria specified in Department of Health and Human Services code § 46.104(d)(2).

There were no medical records or archived samples.

We have included an Ethics Statement subsection within Methods.

We have reviewed the reference list to ensure that it is complete and correct

---

## [Decision Letter · Decision Letter 1]

2 Aug 2023

PONE-D-23-00691R1American Public Opinion on Artificial Intelligence in HealthcarePLOS ONE

Dear Dr. Rojahn,

Thank you for submitting your manuscript to PLOS ONE. After careful consideration, we feel that it has merit but does not fully meet PLOS ONE’s publication criteria as it currently stands. Therefore, we invite you to submit a revised version of the manuscript that addresses the points raised during the review process.

We look forward to receiving your revised manuscript.

Kind regards,

Ali B. Mahmoud, Ph.D.

Academic Editor

PLOS ONE

Journal Requirements:

Reviewers' comments:

Reviewer's Responses to Questions

**Comments to the Author**

1. If the authors have adequately addressed your comments raised in a previous round of review and you feel that this manuscript is now acceptable for publication, you may indicate that here to bypass the “Comments to the Author” section, enter your conflict of interest statement in the “Confidential to Editor” section, and submit your "Accept" recommendation.

Reviewer #2: All comments have been addressed

Reviewer #3: All comments have been addressed

2. Is the manuscript technically sound, and do the data support the conclusions?

Reviewer #2: Yes

Reviewer #3: Yes

3. Has the statistical analysis been performed appropriately and rigorously? 

Reviewer #2: Yes

Reviewer #3: Yes

4. Have the authors made all data underlying the findings in their manuscript fully available?

Reviewer #2: Yes

Reviewer #3: Yes

5. Is the manuscript presented in an intelligible fashion and written in standard English?

Reviewer #2: Yes

Reviewer #3: No

6. Review Comments to the Author

Reviewer #2: This is a valuable analysis of an issue that couldn't be more timely and relevant. It is very readable, too.

Reviewer #3: There are a couple of English errors:

Abstract, line 3:

high expectations for the future of medical AI does exist .. .... do exist

Page 13 line 2:

previous data does not exist  .. do not exist

Page 13 line 8:

interpreted by the subjects of those decisions  the word subject is confusing, it seem to indicate the people who made the decisions instead of the people for which the AI made the decisions.

Overall, the results of the questionnaire look somehow trivial and expected. The people who answered the questions are unlikely to have been exposed to a decision on their health made by AI, so with no first hand experience they produced obvious answers. It would have been useful to ask the subjects if they had any personal experience with an AI application used for health. A more rich set of questions, perhaps including some short videos of doctors and of AI systems, would likely have elicited different answers. The authors could have asked a few additional questions to be able to refine their analysis of the answers and get more insights on the opinions of the people who took the survey.

What is interesting is that despite the fact people think of AI as being less empathetic than a person, some large language models are being used to train the doctors to be more empathetic when they talk with their patients.

The paper does not say when the survey was conducted, which is important since AI advances are reported almost daily by the press, and that can change the opinions of the general public. Also, the number of subjects is relatively small to make inferences about the US population. No information is given on how many people were asked to submit the survey.

7. PLOS authors have the option to publish the peer review history of their article (what does this mean?). If published, this will include your full peer review and any attached files.

Reviewer #2: No

Reviewer #3: No

---

## [Author Response · Author response to Decision Letter 1]

3 Sep 2023

September 3, 2023

Dr. Mahmoud, 

Thank you very much for the reviews and request to revise. We are submitting today a revised version and this response letter addressing all the reviewer points and journal requirements.

Thank you for your time and attention.

Sincerely,

Jessica Rojahn

Please review your reference list to ensure that it is complete and correct. Any changes to the reference list should be mentioned in the rebuttal letter that accompanies your revised manuscript.

We have reviewed the reference list to ensure that it is complete and correct. 

We removed a duplicate reference from the reference list and updated the list order. 

There are a couple of English errors:

Abstract, line 3:

high expectations for the future of medical AI does exist .. .... do exist.

We have corrected this error. 

Page 13 line 2:

previous data does not exist  .. do not exist

We have corrected this error. 

Page 13 line 8:

interpreted by the subjects of those decisions  the word subject is confusing, it seem to indicate the people who made the decisions instead of the people for which the AI made the decisions.

We have reworded this sentence to make it more clear. 

Overall, the results of the questionnaire look somehow trivial and expected. The people who answered the questions are unlikely to have been exposed to a decision on their health made by AI, so with no first hand experience they produced obvious answers. 

We strongly disagree with these unfounded assertions. The reviewer provides no evidence the “results of the questionnaire” are either trivial or expected. The reviewer does not mention any specific result, nor any reason the variable responses to the multiple items were “obvious.”

We have made no revisions in response to these comments, because it is impossible to retroactively surprise the reviewer.

It would have been useful to ask the subjects if they had any personal experience with an AI application used for health. A more rich set of questions, perhaps including some short videos of doctors and of AI systems, would likely have elicited different answers. The authors could have asked a few additional questions to be able to refine their analysis of the answers and get more insights on the opinions of the people who took the survey.

The reviewer has a different instrument in mind. It is not clear to us that short videos of doctors and short videos of AI systems would add any value to the current instrument. One can always imagine additional items on any survey. Here we were constrained to exactly 10 items due to the survey platform. 

The assertion that videos “would likely have elicited different answers” is an empirically testable proposition. It is outside the scope of the current work, and the authors doubt the reviewer’s assertion.

We have made no revisions in response to these comments.

What is interesting is that despite the fact people think of AI as being less empathetic than a person, some large language models are being used to train the doctors to be more empathetic when they talk with their patients.

This is a potentially interesting aside. We would be interested in a reference to whatever work the reviewer is thinking of. However, none of the current results address perceptions of empathy. 

We have made no revisions in response to these comments.

The paper does not say when the survey was conducted, which is important since AI advances are reported almost daily by the press, and that can change the opinions of the general public. 

We thank the reviewer for pointing out this oversight. The data was collected in March 2021. We have added this information to the Methods section. 

Also, the number of subjects is relatively small to make inferences about the US population. 

When statistics are properly implemented, the phrase “the number of subjects is relatively small to make inferences” is devoid of meaning. We have included confidence intervals on every estimate and a p-value for each statistical test. Of course, the common pitfalls of sample-to-population inference apply to the current results as they do to every published quantitative result.

A scale to measure attitudes toward AI is described in: Schepman, A., & Rodway, P. (2020). Initial validation of the general attitudes towards Artificial Intelligence Scale. Computers in human behavior reports, 1, 100014. This scale is published in a peer-reviewed journal and has been cited 101 times. The validation in Schepman & Rodway (2020) is based on 100 respondents. In the current results, we surveyed 203 respondents.

We have made no revisions in response to this comment.

Per the Editor’s request, we have submitted this response letter, a marked-up copy of the manuscript that highlights changes made, and an unmarked version of the revised paper without tracked changes.

We look forward to a decision from the editor following our revisions and this response.

---

## [Decision Letter · Decision Letter 2]

25 Oct 2023

American Public Opinion on Artificial Intelligence in Healthcare

PONE-D-23-00691R2

Dear Dr. Rojahn,

We’re pleased to inform you that your manuscript has been judged scientifically suitable for publication and will be formally accepted for publication once it meets all outstanding technical requirements.

Kind regards,

Ali B. Mahmoud, Ph.D.

Academic Editor

PLOS ONE

Additional Editor Comments (optional):

Reviewers' comments:

Reviewer's Responses to Questions

**Comments to the Author**

1. If the authors have adequately addressed your comments raised in a previous round of review and you feel that this manuscript is now acceptable for publication, you may indicate that here to bypass the “Comments to the Author” section, enter your conflict of interest statement in the “Confidential to Editor” section, and submit your "Accept" recommendation.

Reviewer #3: All comments have been addressed

2. Is the manuscript technically sound, and do the data support the conclusions?

Reviewer #3: Yes

3. Has the statistical analysis been performed appropriately and rigorously? 

Reviewer #3: I Don't Know

4. Have the authors made all data underlying the findings in their manuscript fully available?

Reviewer #3: Yes

5. Is the manuscript presented in an intelligible fashion and written in standard English?

Reviewer #3: Yes

6. Review Comments to the Author

Reviewer #3: To answer a question you raised about a comment I made. I was referring to the fact that towards the end of April 2023,

there were a lot of sites reporting that doctors were using ChatGPT.

Example: The New York Times, https://www.nytimes.com › 2023/06/12, When Doctors Use a Chatbot to Improve Their Bedside Manner

Example: "More empathy from ChatGPT? Medical chatbot beats doctors in bedside manner" A panel of licensed health care professionals preferred ChatGPT’s responses nearly 80% of the time and rated the chatbot’s responses as higher quality and more empathetic, in a new JAMA study. By By Alan Goforth | May 08, 2023 at 11:09 AM

Those were not scientific studies, but some scientific studies have appeared more recently.

7. PLOS authors have the option to publish the peer review history of their article (what does this mean?). If published, this will include your full peer review and any attached files.

Reviewer #3: No

---

## [Editor Report · Acceptance letter]

31 Oct 2023

PONE-D-23-00691R2 

American Public Opinion on Artificial Intelligence in Healthcare 

Dear Dr. Rojahn:

I'm pleased to inform you that your manuscript has been deemed suitable for publication in PLOS ONE. Congratulations! Your manuscript is now with our production department. 

Kind regards, 

on behalf of

Dr. Ali B. Mahmoud 

Academic Editor

PLOS ONE